# An Integrated Bioinformatics Study of a Novel Niclosamide Derivative, NSC765689, a Potential *GSK3β*/*β-Catenin*/*STAT3*/*CD44* Suppressor with Anti-Glioblastoma Properties

**DOI:** 10.3390/ijms22052464

**Published:** 2021-02-28

**Authors:** Ntlotlang Mokgautsi, Ya-Ting Wen, Bashir Lawal, Harshita Khedkar, Maryam Rachmawati Sumitra, Alexander T. H. Wu, Hsu-Shan Huang

**Affiliations:** 1PhD Program for Cancer Molecular Biology and Drug Discovery, College of Medical Science and Technology, Taipei Medical University and Academia Sinica, Taipei 11031, Taiwan; d621108006@tmu.edu.tw (N.M.); d621108004@tmu.edu.tw (B.L.); d621108005@tmu.edu.tw (H.K.); maryamrachma60@gmail.com (M.R.S.); 2Graduate Institute for Cancer Biology and Drug Discovery, College of Medical Science and Technology, Taipei Medical University, Taipei 11031, Taiwan; 3Department of Neurosurgery, Taipei Medical University-Wan Fang Hospital, Taipei 11031, Taiwan; 98142@w.tmu.edu.tw; 4TMU Research Center of Cancer Translational Medicine, Taipei Medical University, Taipei 11031, Taiwan; 5The PhD Program of Translational Medicine, College of Science and Technology, Taipei Medical University, Taipei 11031, Taiwan; 6Clinical Research Center, Taipei Medical University Hospital, Taipei Medical University, Taipei 11031, Taiwan; 7Graduate Institute of Medical Sciences, National Defense Medical Center, Taipei 11490, Taiwan; 8School of Pharmacy, National Defense Medical Center, Taipei 11490, Taiwan; 9PhD Program in Drug Discovery and Development Industry, College of Pharmacy, Taipei Medical University, Taipei 11031, Taiwan

**Keywords:** glioblastoma multiforme (GBM), tumor microenvironment (TME), glioma stem cell (GSC), stemness, drug resistance, in silico molecular docking, *miR-135b*

## Abstract

Despite management efforts with standard surgery, radiation, and chemotherapy, glioblastoma multiform (GBM) remains resistant to treatment, which leads to tumor recurrence due to glioma stem cells (GSCs) and therapy resistance. In this study, we used random computer-based prediction and target identification to assess activities of our newly synthesized niclosamide-derived compound, NSC765689, to target GBM oncogenic signaling. Using target prediction analyses, we identified *glycogen synthase kinase 3β* (*GSK3β*), *β-Catenin*, signal transducer and activator of transcription 3 (*STAT3*), and cluster of differentiation 44 (*CD44*) as potential druggable candidates of NSC765689. The above-mentioned signaling pathways were also predicted to be overexpressed in GBM tumor samples compared to adjacent normal samples. In addition, using bioinformatics tools, we also identified microRNA (miR)-135b as one of the most suppressed microRNAs in GBM samples, which was reported to be upregulated through inhibition of *GSK3β*, and subsequently suppresses GBM tumorigenic properties and stemness. We further performed in silico molecular docking of NSC765689 with GBM oncogenes; *GSK3β*, *β-Catenin*, and *STAT3*, and the stem cell marker, *CD44*, to predict protein-ligand interactions. The results indicated that NSC765689 exhibited stronger binding affinities compared to its predecessor, LCC09, which was recently published by our laboratory, and was proven to inhibit GBM stemness and resistance. Moreover, we used available US National Cancer Institute (NCI) 60 human tumor cell lines to screen in vitro anticancer effects, including the anti-proliferative and cytotoxic activities of NSC765689 against GBM cells, and 50% cell growth inhibition (GI_50_) values ranged 0.23~5.13 μM. In summary, using computer-based predictions and target identification revealed that NSC765689 may be a potential pharmacological lead compound which can regulate GBM oncogene (*GSK3β/β-Catenin*/*STAT3*/*CD44*) signaling and upregulate the *miR-135b* tumor suppressor. Therefore, further in vitro and in vivo investigations will be performed to validate the efficacy of NSC765689 as a novel potential GBM therapeutic.

## 1. Introduction

Glioblastoma multiforme (GBM) is a heterogeneous and extremely aggressive intracranial cancer, commonly diagnosed in the adult brain. It is the most lethal malignancy of the central nervous system (CNS) [1], and its aggressive character is associated with poor prognoses despite the efforts of standard surgery, radiation, and chemotherapy [2], in which fewer than 4.3% of late-stage patients reach the 5-year survival mark [3]. GBM represents almost 57% of all gliomas and approximately 48% of all primary tumors of the CNS [4]. The CNS and GBM tumor microenvironment (TME) play significant roles in regulating the effectiveness of antitumor responses and activation of immune responses towards cancer cells [5]. GBM is characterized by vascular proliferation, inflammation, intratumoral heterogeneity (such as necrosis), an invasive mechanism, and an ability to escape immune responses [6]. Moreover, there are also complex interactions between the tumor and the TME [7]. In addition to factors within the TME, GBM tumor cells employ different mechanisms to exert their cancerous potential in patients, and this ultimately leads to poor prognoses [8]. The TME contains a variety of non-malignant cell types, including surrounding stromal cells, such as endothelial cells, immune cells, and fibroblasts [9].

According to recent studies, the most profuse cells within the TME are cancer-associated fibroblasts (CAFs), which progressively influence biological properties of GBM, such as tumor development, and are linked to late cancer stages. CAFs also play a significant role in the development of glioma stem cells (GSCs) by regulating the TME [10]. GSCs, also known as self-renewing cells, were reported to drive tumor progression and recurrence, and are resistant to therapeutic interventions [11]. GSCs can be isolated in a patient’s tumor using single surface markers including cluster of differentiation 44 (*CD44*) among others [12,13]. Glioma cells secrete immunostimulatory molecules which enhance communication among circulating stem cells, astrocytes, immune cells, and the TME, leading to the production of cytokines and growth factors. This weakens the immune response and promotes tumor development and progression as well as treatment resistance [7,14]. GBM proliferation, migration, and adhesion were shown to be promoted by *β-Catenin*, a component of the canonical *Wnt* pathway, which is responsible for regulating cell-cell adhesion and transcription of genes [15]. The *β-Catenin expression* level was demonstrated to be upregulated in GBM cancer cells, leading to tumorigenesis and progression [16]. Previous studies showed that patients with overexpression of *N-Myc* downregulated gene 1 (NDRG1) in GBM tumor cells exhibited better prognoses, and its downregulation increased the expression level of *glycogen synthase kinase 3β* (*GSK3β*), which promotes cell proliferation [17]. *GSK3β* is a threonine kinase, which is associated with a variety of physiological cell processes including proliferation and cell death.

In clinical settings, *GSK3β* was reported to be associated with therapeutic resistance in glioblastoma cancer patients [18]. However, its role in GBM still remains to be fully explored [19]. In addition, others also reported that inhibition of *NDRG1* modulates the epithelial-to-mesenchymal transition (EMT), which is a major regulator of the *Wnt/β-Catenin pathway* [20]. Signal transducer and activator of transcription 3 (*STAT3*) is a cytoplasmic transcription factor which regulates several cellular processes, such as cell differentiation, migration, and survival [21]. When activated, *STAT3* regulates the GBM mesenchymal transition and tumor progression [22,23]. As a result, targeting the *STAT3* signaling axis could prove to be significant for managing cancer. In GBM, the *STAT3* pathway was shown to interact with *GSK3β* [24]. Furthermore, accumulating evidence has identified a subset of cancer cells which exhibits glioblastoma stem-like cell (GSC) properties; this subpopulation of cells is highly resistant to treatment [25]. Therefore, there is a great need for novel and effective therapeutic agents for GBM. Mechanisms leading to GBM metastasis and aggressiveness still remain elusive; however, the molecular identification of GBM has revealed a novel therapeutic strategy by specifically targeting oncogenes or microRNAs (miRNAs; miRs) [26,27]. Studies showed that *miR-135b* is significantly downregulated in GSCs compared to normal stem cells [28]. Additionally, reports showed that by targeting *GSK3β*, *miR-135b* expression increases and consequently inhibits GSC proliferation, migration, and invasion [29].

In this study, we used target prediction analyses to identify *GSK3β*, *β-Catenin*, *STAT3*, *and CD44* as potential druggable candidates of NSC765689. These signaling pathways were also predicted to be overexpressed in GBM tumor samples compared to adjacent normal samples. In addition, using bioinformatics tools, we also identified *miR-135b* as one of the most suppressed miRNAs in GBM samples, which was reported to be upregulated through inhibition of its target gene *GSK3β*, and subsequently suppress GBM tumorigenic properties and stemness. We further performed in silico molecular docking of NSC765689 with the GBM oncogenes; *GSK3β*, *β-Catenin*, *and STAT3*, and the stem cell marker of *CD44*, to predict protein-ligand interactions, and results indicated that NSC765689 exhibited stronger binding affinities compared to its predecessor, LCC09, which was recently published by our laboratory, and proved to inhibit GBM stemness and resistance. Moreover, we used available US National Cancer Institute (NCI) 60 human tumor cell lines to screen for in vitro anticancer effects, including cytotoxic activities of NSC765689 against GBM cells.

## 2. Results

### 2.1. NSC765689 Successfully Meets Required Drug-Likeness Criteria

NSC765689 is a close ring, and its predecessor LCC09 an open ring, both of these compounds are derived from niclosamine (Figure 1A) [30]. To identify potential drug candidates in the early stage of drug discovery and development, we applied criteria specified for the drug-likeness concept [31,32]. By utilizing the BOILED-egg prediction, we evaluated gastrointestinal absorption (GIA) and blood-brain-barrier (BBB) permeability of the compound, and results showed that NSC765689 had the highest probability of having good GIA and BBB permeation Bioavailability radar (Figure 1B), this enabled an evaluation of the drug-likeness of NSC765689. All six physicochemical properties outlined by bioavailability radar showed that NSC765689 passed the minimum requirements of drug-likeness (Figure 1B and accompanying table), based on the molecular weight of the compound, lipophilicity, polarity, solubility, saturation, flexibility, and required criterion of log *S* (ESOL) = 0~6, which is accepted as good solubility in the body. However, our compound did not meet the required log *S* (ESOL) criterion with a value of -5.34. Furthermore, pharmacokinetics, drug-likeness, and medicinal chemical properties of NSC765689 (Table 1) indicated that NSC765689 had good synthetic accessibility of 3.27 and met criteria for the Lipinski (Pfizer) rule-of-five, the Ghose, Veber (GSK), Egan (Pharmacia), and muegge for drug likeness and drug discovery. The bioavailability of the compound based on GIA indicated a score of 0.55 (55%) which indicates acceptable pharmacokinetic properties.

### 2.2. NSC765689 Exhibits Similar Anticancer Fingerprint with NCI Synthetic Compounds and Standard Agents

DTP-COMPARE analysis indicated that NSC765689 share similar anticancer fingerprints with a number of NCI synthetic compounds and NCI standard agents. The Pearson’s correlation coefficient (r) and common cell lines count cell counts (CCLC) of the top 15 NCI correlated standard agents are shown in (Table 1) All NCI synthetic compounds with NSC765689 similar anticancer fingerprints are small molecules with the molecular weight (MW; 177.18~470.17g/mol).

### 2.3. GSK3β/β-Catenin/STAT3/CD44 Are Potential Druggable Candidates for NSC765689

Using a computer-based target prediction tool (http://www.swisstargetprediction.ch/ (accessed on 20 January 2021)) with NSC765689 as the target compound, we identified several targetable proteins, including kinases, enzymes, transcription factors, proteases, and other unclassified proteins (Appendix A). Our compound was also shown to target oncogenes such as *Janus kinase 2/3* (*JAK2/3*), *epidermal growth factor receptor* (*EGFR*), *GSK3β*, *mitogen-activated protein kinase* (*MAPK*), *phosphatidylinositol 3-kinase α* (*PI3KCA*), *STAT3*, and *cyclin-dependent kinases* (*CDKs*) as shown below (Table 2, Appendix A).

### 2.4. Increased GSK3β/β-Catenin/STAT3 Expressions Are Associated with Poor Prognoses

To test our hypothesis, we first utilized bioinformatics analyses of the public databases and found that using the online bioinformatics tool GEPIA2 (http://gepia2.cancer-pku.cn (accessed on 20 January 2021)) with the default settings, messenger (m)RNA levels of *GSK3β*, *β-Catenin*, *STAT3*, *and CD44* were highly expressed in tumor tissues of patients with GBM compared to normal tissues with significant *p* values (<0.05), and the correlational analysis showed positive correlations of *STAT3* with *GSK3β*, *CTNNB1* (*β-Catenin*) with *GSK3β*, *STAT3* with *CD44*, and *CTNNB1* with *CD44* in GBM patients (*r* = 0.28~0.58) (Figure 2). In further analyses using the SurvExpress online tool with default settings, (http://bioinformatica.mty.itesm.mx:8080/Biomatec/SurvivaX.jsp (accessed on 20 January 2021)), higher *GSK3β/β-Catenin*/*STAT3* expression levels were associated with significantly shorter survival times compared to patients with lower *GSK3β/β-Catenin*/*STAT3* expression levels. To explore further, when all four genes associated with immune surveillance escape, viz., *GSK3β*/*β-Catenin*/*STAT3*/*CD44*, were taken into account, patients with high expression levels of these four genes had significantly shorter survival times, which were even more significant than when just considering *GSK3β*/*β-Catenin*/*STAT3* signaling, suggesting the predictive power of this signature (Figure 3). In additional analyses, we used the STRING database (https://string-db.org/cgi/ (accessed on 20 January 2021)) under high-confidence (with a minimal interaction score of 0.700) predictions to predict protein-protein interactions (PPIs) among these proteins, and results indicated gene expression correlation networks of *GSK3β* with *CTNNB1*, *GSK3β* with *STAT3 CD44* with *STAT3*, and *CD44* with *CTNNB1.* Correlation networks based on curated data and experimentally validated data existed among all four of these oncogenes. The network has four initial proteins formed and was further expanded by an additional 30 nodes/protein, by using the “more” key on the STRING interface. Furthermore, a PIN enrichment *p* value of <1.0 × 10^−16^ was obtained from the network, with a clustering coefficient of 0.711. The table shows all other interacting proteins with *GSK3β*/*β-Catenin*/*STAT3*/*CD44*, and the confidence cutoff value representing interaction links was adjusted to 0.900 as the highest scoring link (Figure 4A, accompanying table). From the above results we analyzed the gene ontology biological processes and related pathways, (Figure 4B) shows the top 10 most highly associated biological processes in the *GSK3β*/*β-Catenin*/*STAT3*/*CD44* networks included glial cell differentiation, response to drug, canonical *Wnt* signaling, *MAPK* cascade, brain development, innate immune response, apoptosis processes, positive regulation of threonine kinase, and (Figure 4C) Kyoto Encyclopedia of Genes and Genomes (KEGG) pathways that were linked with *GSK3β*/*β-Catenin*/*STAT3*/*CD44* networks included AMPK signaling, glioma, cellular senescence, microRNAs in cancer, *MTOR*, *PI3K/AKT*, *JAK-STAT*, *Wnt* signaling and pathways in cancer.

### 2.5. Increased Expressions of the GSK3β/β-Catenin/STAT3/CD44 Signatures and Reduced miR-135b Are Associated with Poor Prognoses of GBM Patients

GSK3B was reported to promote GBM cancer development and progression and has emerged as a target for drug development [33]. Subsequently, we used the online software TargetScan (http://www.targetscan.org/vert (accessed on 20 January 2021)) to search for potential target genes of *miR-135b* -5p. We identified an *miR-135b -5p* site in the 3′ untranslated region (UTR) of *GSK3β* mRNA, thereby predicting *GSK3β* to be one of the potential genes. Using the miRandola database (http://mirandola.iit.cnr.it/ (accessed on 20 January 2021)), we identified *miR-135b -5p* to be significantly downregulated in GBM cancer patients compared to normal cancer samples of “black columns” (Figure 5).

### 2.6. In Silico Molecular Docking Showed Putative Binding of NSC765689 with GSK3β/β-Catenin/STAT3/CD44

We further performed in silico molecular docking analyses to assess the potential inhibitory effects of NSC765689 compared to its predecessor, LCC09, on oncogene markers including *GSK3β*, *β-Catenin*, and *STAT3* and the stem cell marker, *CD44*. Results of protein-ligand interactions obtained through AutoDock Vina showed that NSC765689 displayed the highest respective binding energies of −10.3, −8.2, −9.2, and −8.0 kcal/mol with the *GSK3β*, *β-Catenin*, *STAT3*, *and CD44* genes, compared to its predecessor, LCC09, which displayed slightly lower binding affinity values of (−9.3, −7.9, −8.8, and −7.7) kcal/mol, respectively. Visualization by Pymol and interpretation using the discovery studio results revealed that the NSC765689/*GSK3β* complex and LCC09/*GSK3β* complex showed similar interactions by conventional hydrogen (H)-bonds with Arg144 and THR138, by Pi-anion interactions with LYS85, and by Pi-Pi stacked interactions with ASP200, and it is further stabilized by Pi-alkyl interactions with VAL70, and also by Van der Waal interactions with SER66 as displayed in their binding pockets. Furthermore, the NSC765689/*β-Catenin* complex interacted with *β-Catenin* by H-bonds with Arg457, THR418, ASN415, ASN387, and TRP383, and was further sustained by Pi-anion interactions with ASP459 and by Pi-alkyl interactions with ILE414 as shown in their binding pockets. The NSC765689/*STAT3* complex and LCC09/*STAT3* complex exhibited the same interactions by conventional H-bond interactions with TRP37 and GLU63 in the binding pockets. The NSC765689/*CD44* complex was stabilized by conventional H-bond interactions with ASN94 and GLN113 and further sustained by Pi-anion interactions with ARG90 at their binding sites (Figure 6, accompanying table) and (Figure 7, accompanying table). Moreover, we compared docking results of *GSK3β*/*β-Catenin*/*STAT3*/*CD44* in complex with LCC09 and NSC765689 to *GSK3β*/*β-Catenin*/*STAT3* docking with standard inhibitors of AZD1080, cardamonin, and cryptotanshinone. Interestingly, results showed that NSC765689 exhibited the highest binding energies with *GSK3β/β-Catenin*/*STAT3*/*CD44* of (−10.3, −8.2, −9.2, and −8.0) kcal/mol, respectively, compared to standard inhibitors with binding energies of (−8.5, −6.0, and −7.0) kcal/mol with *GSK3β/β-Catenin*/*STAT3*, respectively (Figure 8) and accompanying table). Collectively, these structural simulations predicted NSC765689 to be a multi-target inhibitor with high confidence.

### 2.7. NSC765689 Exhibits Cytotoxic Activities Obtained from Single High-Dose Testing of 60 Human Cancer Cell Lines (NCI)

We investigated the 60 human tumor cell line available from the US NCI to screen for anticancer effects, including cytotoxicity of NSC765689 toward multiple cell lines of leukemia, non-small cell lung cancer (NSCLC), colon cancer, central nervous system (CNS) cancers, melanomas, ovarian cancers, renal cancers, prostate cancer, and breast cancer based on the developed therapeutic program of the US NCI [34]. Primarily, we evaluated the anti-proliferation and cytotoxic effects of NSC765689 against the NCI-60 cell lines, and we discovered that NSC765689 showed anticancer activities against these cancer cell lines (Figure 9). The cytotoxic activities of NSC765689 were shown after an initial single dose (10 μM), through which the effects of the compound are represented by the percentage growth modified by the treatment (Table 3). Melanoma cell lines were more sensitive to NSC765689, and the activities of the drug growth inhibition (GI) were 67.21%, 29.37%, 28.31% for the SK-MEL-5, MALME-3M, and SK-MEL-2 cell lines, respectively, followed by 39.25%, 34.46%, and 28.76% GI for the HL-60 (TB), MOLT-4, and RPMI-8226 leukemia cell lines, respectively. Renal cancer GI was 35.34% for the A498 and 23.92% for the UO-31 cell lines. The MDA-MB-231/ATCC breast cancer cell line was also sensitive to the drug at 27.93% as was the MDA-MB-468 cell line with −18.67% GI. A CNS cancer cell line exhibited −19.35% GI, followed by NSCLC, ovarian cancer, colon cancer, and prostate cancer which were less sensitive to NSC765689 compared to the above-mentioned cancer types, with the GI percentages of −13.04%, −7.61%, and −4.67%, 0% respectively, however the compound also exhibited satisfactory anti-proliferative effects on all the above mentioned cancer cells lines (Figure 10). The initial single dose of NSC765689 of 10 μM exhibited anticancer effects against multiple cancer cell lines, and proved that more investigation of the dose-dependent effects would be worthwhile. For the purpose of this study, however, we mainly focused on the CNS cancer data, since our main target cancer type was GBM, which was reported to account for approximately 45% of all primary CNS tumors [35].

### 2.8. NSC765689 Exhibits Dose-Dependent Anticancer Activities against NCI-60 Human Cancer Cell Lines

Since NSC765689 exhibited satisfactory anticancer activities against the 60 NCI cancer cell lines, at an initial single dose of 10 μM, further exploration was performed. However, this time, we focused on the CNS results as a way of targeting GBM cancer. Results showed that by treating cell lines with five doses of NSC765689, CNS cell lines (SF-268, SF-295, U251, and SNB-75) were more sensitive in dose-dependent manners (Figure 10). The 50% GI (GI_50_) concentrations of NSC765689 against most of the NCI-60 human cell lines ranged 0.23~5.13 μM; the most sensitive cell line was a melanoma (SK-MEL-5) cell line which showed a small sub-micromolar GI_50_ value of 0.23 μM, followed breast cancer (BT-549), leukemia (MOLT-4), NSCLC (NCI-H46-), ovarian cancer (OVCAR-4), renal cancer (A489), and CNS (SNB-75) with values of 0.29, 0.30, 0.36, 0.38, 0.40 and 0.61 μM respectively. The lowest GI activity was for the COLO 205 colon cancer cell line with a value of 5.13 μM (Figure 9 and Figure 10).

## 3. Discussion

GBM is the leading type of glial tumor and most aggressive primary brain tumor in adults [36]. Despite considerable advances in treatment modalities including surgery, radiation, and chemotherapy, GBM patients eventually develop resistance to therapy, and the overall survival (OS) rate remains under 2 years, due to the heterogeneous character of GBM [11]. Therapeutic resistance is associated with the development of glioma stem cells (GSCs), which were demonstrated to be capable of withstanding treatment interventions [37]. Emerging studies demonstrated several challenges in treating GBM, including intertumoral heterogeneity, insufficient drug efficacy, and poor drug permeability across the BBB [38]. In addition, GBM cells seldom metastasize; however, cells move from the tumor to penetrate neighboring brain tissues, hence rendering treatment inadequate [6,39]. This necessitates the need to develop novel treatments with the ability to target GBM cellular heterogeneity and GSC therapeutic resistance. In the current study, we demonstrated the anticancer activities of a novel compound NSC765689 against a panel of 60 human tumor multiple cell lines of the US national cancer institute (NCI). NSC765989 (closed ring), is a niclosamide-derived compound synthesized in our laboratory. Based on an initial single dose of 10 μM, the antiproliferative and cytotoxic effects of the tumor cell lines represented by the percentage growth showed the malignant cells to be sensitive to NSC765689, with GI_50_ values ranging 0.23~5.13 μM against most of the HCI-60 human cell lines. Melanoma cell lines were more sensitive, followed breast cancer, leukemia, NSCLC, ovarian cancer, renal cancer, and CNS and prostate cancer cell lines which were less sensitive to NSC765689. For the purpose of this study, since we were focusing on GBM, which is the most common malignancy of the CNS [1], We further explored the dose-dependency at five doses on CNS cell lines and found that the SF-268, SF-295, U251, and SNB-75 CNS cell lines were more sensitive to NSC765689 (Figure 10).

The development therapeutics program (DTP) of the NCI has demonstrated that cell lines growth inhibition pattern was common among compounds with the same mechanisms, herein, we applied the DTP-COMPARE analysis to identify other compounds which shares similar patterns of percentage growth inhibition with NSC765689 [40]. The results from the analysis indicated that NSC765689 share similar anticancer mechanisms with a number of NCI standard agent with pearson’s correlation (*r* = 0.25~0.4). The compound also shared the similar anticancer fingerprints with a number of NCI synthetic compounds with (*r* = 0.54~0.59). The Pearson’s correlation coefficient (r) and (CCLC) of the top 15 NCI correlated standard agents are shown in Table 2. All NCI synthetic compounds with NSC765689 similar anticancer fingerprints are small molecules with the molecular weight (MW; 177.18~470.17g/mol). In addition, COMPARE prediction tool and swiss target software predicted *GSK3β*/*β-Catenin*/*STAT3*/*CD44* as target gene for NSC765689. For furthermore analysis, we utilized The Cancer Genome Atlas (TCGA) public database and identified *GSK3β*, *β-Catenin*, *STAT3*, and *CD44* oncogenes to be overexpressed in GBM tumor samples compared to adjacent normal tissue groups, and this was associated with poor cancer prognoses (Figure 2). Moreover, we explored interactions among these oncogenes using the STRING analytical tool. PPI networks (PINs) were previously used as a blueprint to analyze different biological activities, including identifying signaling pathways and mechanisms through which cellular processes are activated, such as cell development and immunity [41,42]. As anticipated, we found a clustering network of *GSK3β*/*β-Catenin*/*STAT3*/*CD44* and other proteins displayed in the string interface, which were related to the cell cycle. A PIN enrichment *p* value < 1.0 × 10^−16^ was obtained from the network, with a clustering coefficient of 0.711, and the confidence cutoff value representing the interaction links was adjusted to 0.900 as the highest scoring link (Figure 4). Accumulating reports have demonstrated that *GSK3β* is highly expressed in GBM compared to non-malignant brain tissues, and plays a significant role as a promoter of neoplastic GBM phenotypes, making it a potential biomarker for targeted therapy [19,43,44]. Therefore, STRING analytical results showed that proteins which interacted the most with *GSK3β* in the network were *STAT3*, *AKT*, *β-Catenin*, *SNAI1*, and *adenomatous polyposis coli* (*APC*), thus supporting our hypothesis.

There are several other factors that contribute to GBM tumorigenesis, progression, and therapeutic resistance, including microRNAs (miRs) [45]. miRs are approximately 22-nucleotide-long non-coding RNAs, which play significant roles in modulating gene expressions after DNA transcription [46]. Recent studies showed another important function of miRs as a tumor suppressor depending on their target genes [47]. Using miR target prediction online software, we identified significantly lower expression of the *miR-135b* tumor suppressor in GBM tumor cohorts compared to normal samples, with the fold change (LogFC = −0.733) and (*p*-value < 0.001). In addition, we discovered that *miR-135b* is downregulated in GBM patients (Figure 5A,B), and according to recent studies, it plays a pivotal role in GBM progression, and its downregulation is also associated GBM stemness [29]. This suggests that restoration of *miR-135b* could be important strategy to reduce tumorigenesis and cancer recurrence. Since miRs function as tumor suppressors depending on their target gene, we further explored this aspect with an miR target prediction tool, and identified the *miR-135b-5p* site on the 3′UTR region of *GSK3β*, which suggests that *GSK3β* may be a target gene for *miR-135b-5P* (Figure 5C). Thus we suggest further in vitro and in vivo investigations into the potential roles of NSC765689 in targeting *GSK3β* as a way of increasing the *miR-135b* expression level in GBM patients.

In our previous study, one of our small molecule, LCC-09, was shown to inhibit GBM stem cell-related oncogenic activities and ultimately increased the expression level of miR-34a in GBM cells both in vitro and in vivo [37]. From our recent findings, we rationalized that this open-ring niclosamide-derived compound (LCC-09) could be further optimized; thus, we synthesized the closed-ring niclosamide small-molecule NSC765689, which might yield better results, as close ring structures has been proven to be more stable [30,48]. Since *GSK3β/β-Catenin*/*STAT3* showed to be potential NSC765689 druggable targets, we performed in silico molecular docking to analyze ligand-protein interactions. Interestingly, we discovered that NSC765689 was able to bind to *GSK3β/β-Catenin*/*STAT3*/*CD44*, and exhibited stronger binding affinities as compared to its predecessor LCC09 which displayed slightly lower respective binding affinities (Figure 6 and Figure 7 and accompanying tables). Additionally, we compared docking results of *GSK3β/β-Catenin*/*STAT3*/*CD44* in complexes with LCC09 and NSC765689 by docking with standard inhibitors of *GSK3β/β-Catenin*/*STAT3* of AZD1080, cardamonin, and cryptotanshinone, respectively. Interestingly, results showed that NSC765689 exhibited the highest binding affinities than all the aforementioned inhibitors (Figure 8 and accompanying table). NSC765689 successfully met the physicochemical properties as outlined by the bioavailability radar such as lipophilicity, molecular weight, polarity, solubility, saturation, and flexibility. The pharmacokinetics, drug-likeness, and medicinal chemical properties of NSC765689 indicated that NSC765689 had good synthetic accessibility of 3.27 and met criteria for the Lipinski (Pfizer) rule-of-five, and the Ghose, Veber (GSK), Egan (Pharmacia), and muegge for drug likeness and drug discovery. The bioavailability of the compound based on the GIA showed a score of 0.55 (55%), which indicates acceptable pharmacokinetic properties. Therefore, the efficacy of NSC765689 is worthy of further investigation in both in vitro and in vivo studies, which are currently in progress in our laboratory.

## 4. Materials and Methods

### 4.1. Pharmacokinetics, Drug-Likeness, and Medicinal Chemical Analyses

The drug-likeness, medicinal chemistry, and pharmacokinetics investigation examined the adsorption, distribution, metabolism, excretion, and toxicity (ADMET) properties of NSC765689 as analyzed using SwissADME software developed by the Swiss Institute of Bioinformatics (http://www.swissadme.ch (accessed on 20 January 2021)) [49]. The drug-likeness properties were analyzed in terms of Ghose (Amgen), Egan (Pharmacia), Veber (GSK), and more importantly, the Lipinski (Pfizer) rule-of-five: cLogP, molecular mass, hydrogen acceptor, hydrogen donor, and molar refractive index [50] for drug likeness and drug discovery. The Abbot Bioavailability Score was calculated based on the probability of the compound to have at least 0.1 (10%) oral bioavailability in rats or measurable Caco-2 permeability [51], while gastrointestinal absorption and brain penetration properties were analyzed using the Brain Or IntestinaL EstimateD permeation (BOILED-Egg) model [32].

### 4.2. Bioinformatics Predictions

To analyze the overall survival (OS) and disease-free survival (DFS) ratios between cancer cohorts with high and those with low GSK2B/*β-Catenin*/*STAT3*/*CD44* expressions, we utilized Kaplan-Meier survival plots on the comprehensive gene expression database SurvExpress with default settings (http://bioinformatica.Biomatec/SurvivaX.jsp (accessed on 20 January 2021)). For further analysis, we used the STRING database (https://string-db.org/ (accessed on 20 January 2021)) under high confidence (minimal interaction score of 0.700) predictions to predict the PPIs among these proteins, and results indicated gene expression correlation networks of *GSK3β* with *CTNNB1*, *GSK3β* with *STAT3*, and *CD44*, and *STAT3* and *CD44* with *CTNNB1.* We used the UALCAN bioinformatics tool (http://ualcan.path.uab.edu/ (accessed on 20 January 2021)) with default settings for a differential expression analysis of *GSK3β/β-Catenin*/*STAT3*/*CD44* expressions in tumor vs. normal tissues from various cancers in The Cancer Genome Atlas (TCGA) database.

### 4.3. In Silico Molecular Docking Analyses

In silico molecular docking analyses were used to assess the potential inhibitory effects of NSC765689 compared to its predecessor, LCC09, on oncogene markers, including *GSK3β*, *β-Catenin*, and *STAT3*, and the stem cell marker, *CD44*. Results of protein-ligand interactions obtained through AutoDock Vina showed that NSC765689 displayed the highest binding energy values of (−10.3, −8.2, −9.2, and −8.0) kcal/mol with the *GSK3β*, *β-Catenin*, *STAT3*, and *CD44* genes, respectively, compared to its predecessor LCC09 which displayed slightly lower binding affinity values of −9.3, −7.9, −8.8, and −7.7 kcal/mol, respectively. Visualization by Pymol and interpretation using the discovery studio results revealed that the NSC765689/*GSK3β* complex and LCC09/*GSK3β* complex showed similar interactions by conventional hydrogen (H)-bonds with Arg144 and THR138, by Pi-anion interactions with LYS85, and Pi-Pi stacked interactions with ASP200, and were further stabilized by Pi-alkyl interactions with VAL70, and also by Van der Waals interactions with SER66 as displayed in their binding pockets. Furthermore, the NSC765689/*β-Catenin complex* interacted with *β-Catenin by* H-bonds with Arg457, THR418, ASN415, ASN387, and TRP383, and was further sustained by Pi-anion interactions with ASP459 and by Pi-alkyl interactions with ILE414 as shown in their binding pockets. The NSC765689/*STAT3* complex and LCC09/*STAT3* complex exhibited the same interactions by conventional H-bond interactions with TRP37 and GLU63 in their binding pockets. The NSC765689/*CD44* complex was stabilized by conventional H-bond interactions with ASN94 and GLN113 and further sustained by Pi-anion interactions with ARG90 at the binding site (Figure 7 and Figure 8 and accompanying tables). Moreover, we compared docking results of *GSK3β/β-Catenin*/*STAT3*/*CD44* in complex with LCC09 and NSC765689 to *GSK3β/β-Catenin*/*STAT3* docking with standard inhibitors of AZD1080, cardamonin, and cryptotanshinone.

### 4.4. In Vitro Screening Of NSC765689 against the Full NCI-60 Cell Panels of Human Tumor Cell Lines

NSC765689 was submitted to the NCI for screening on its panel of NCI-60 cancer cell lines. The preliminary single-dose screening of the two compounds were conducted against all 60 NCI cell line panels comprised of melanomas, leukemia, CNS cancers, NSCLC, renal cancer, breast cancer, ovarian cancer, and prostate cancer in agreement with the protocol of the NCI (http://dtp.nci.nih.gov (accessed on 20 January 2021)). The NSC765689 compound was tested against 60 cell lines of NCI, with an initial single dose of 10 μM. Results are presented in Figure 10. Melanoma cell lines were more sensitive to NSC765689 and the activities of drug growth inhibition (GI) were 67.21%, 29.37%, and 28.31% for SK-MEL-5, MALME-3M, and SK-MEL-2 cells, respectively. Next were 39.25%, 34.46%, and 28.76% GI levels for the HL-60 (TB), MOLT-4, and RPMI-8226 leukemia cell lines, respectively. Renal cancer GI was 35.34% for the A498 and 23.92% for the UO-31 cell line. The breast cancer cell line of MDA-MB-231/ATCC was also sensitive to the drug at 27.93% as was the MDA-MB-468 cell line with −18.67% GI. A CNS cancer cell line exhibited GI of −19.35%. followed by NSCLC, ovarian cancer, colon cancer, and prostate cancer which were less sensitive to NSC765689 compared to the above-mentioned cancer types, with GI levels of −13.04%, −7.61%, and −4.67% and 0% respectively.

### 4.5. Data Analysis

Pearson’s correlation was used to assess correlations of *GSK3β/β-Catenin*/*STAT3*/*CD44* expressions in GBM. The statistical significance of differentially expressed genes was evaluated using the Wilcoxon test. * *p* < 0.05 was considered statistically significant. The Kaplan-Meier curve was employed to present the patient survival from different cancer cohorts. GI by NSC765689 in the single-dose assay was obtained by subtracting the positive value on the plot from 100, i.e., a value of 40 would mean 60% growth inhibition.

## 5. Conclusions

In summary, overexpression of the *GSK3β*, *β-Catenin*, *STAT3*, and *CD44* oncogenes and downregulation of the *miR-135b* tumor suppressor in GBM tumor samples compared to adjacent normal tissue were associated with poor patient prognoses, stemness, and therapeutic resistance. Recently we published a paper showing that LCC09, a predecessor of our present compound NSC765689, inhibited GBM stem cell-related oncogenesis and ultimately increased the expression level of the tumor suppressor in glioblastoma cells, in vitro and *in vivo*. Herein, we predicted that NSC765689 could successfully target *GSK3β*, *β-Catenin*, *STAT3*, and *CD44* oncogenic signaling, showing its potential role in regulating expressions of these oncogenic signatures in GBM. In addition, through targeting these signaling pathways, will possibly increase the expression of the *miR-135b* tumor suppressor increased in GBM cells. Finally, we reported the in vitro anti-proliferative and cytotoxic effects of the NSC765689 compound in multiple cells line, specifically in GBM cancer cell lines. We are currently conducting in vitro and in vivo studies in our laboratory for further validation of NSC765689 as a GBM therapeutic.

## Figures and Tables

**Figure 1 ijms-22-02464-f001:**
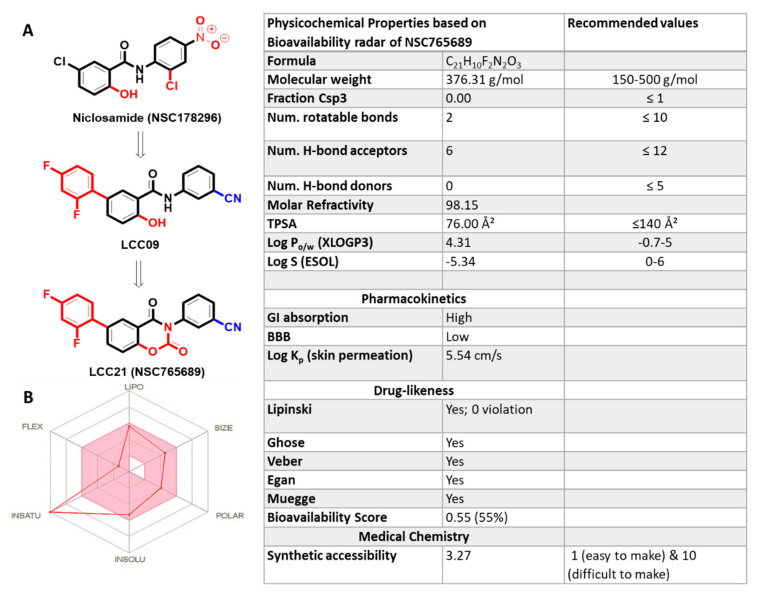
NSC765689 successfully met the required drug-likeness criteria. (**A**) Flow diagram of LCC09 and NSC765689 chemical and structural synthesis with reference to previous studies. (**B**) Bioavailability radar (BA), displaying the six physicochemical properties of absorption of the NSC765689 compound. The pink area represents the optimal range for each property. The accompanying table displays physicochemical properties of lipophilicity of XLOGP3 between −0.7 and 5.0; molecular weight of ≤500 g/mol; polarity of TPSA ≤ 140 Å (angstrom); flexibility of ≤9 rotatable bonds; and solubility of log S between 0 and 6, and saturation sp3 hybridization of ≤0.25.

**Figure 2 ijms-22-02464-f002:**
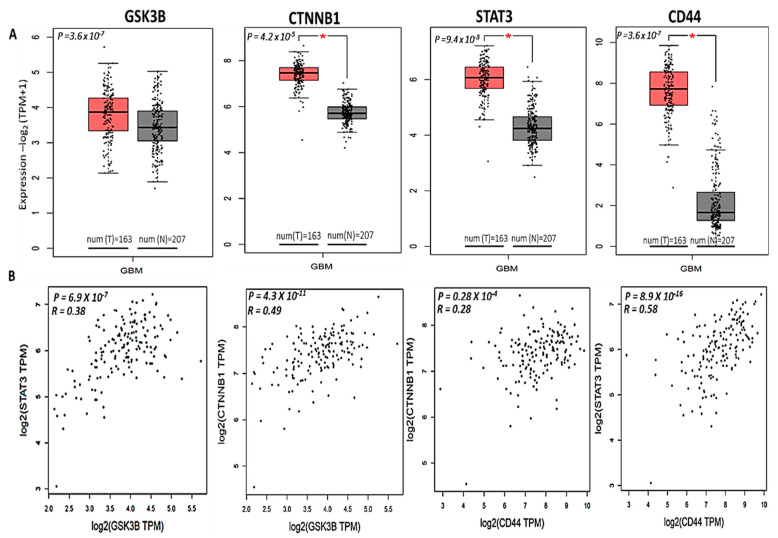
Glycogen synthase kinase 3β (GSK3B)/*β-Catenin*/signal transduction and activator of transcription 3 (*STAT3*)/cluster of differentiation 44 (*CD44*) mRNA levels were highly expressed in glioblastoma multiforme (GBM) tumor cohorts compared to normal samples with significant *p* values (**A**). When all four genes were combined for analysis, there were positive correlations of *STAT3* with *GSK3β*, *CTNNB1* (*β-Catenin*) with *GSK3β*, *STAT3* with *CD44*, and *CTNNB1* with *CD44* in GBM patients (*r* = 0.28~0.58) (**B**). The statistical significance of differentially expressed genes was evaluated using the Wilcoxon test. * *p* < 0.05.

**Figure 3 ijms-22-02464-f003:**
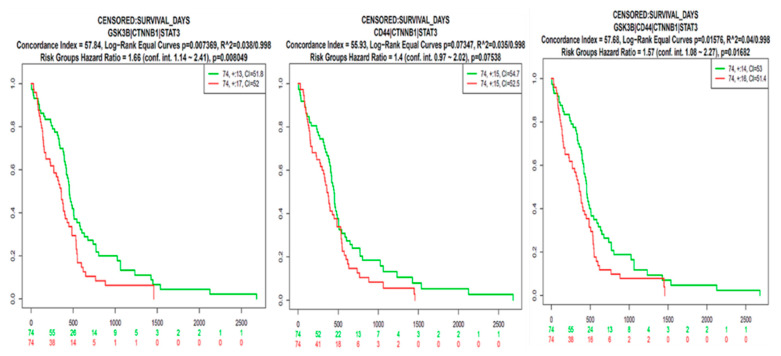
Prediction of shorter survival times using glycogen synthase kinase 3β (*GSK3β*)/*β-Catenin*/signal transduction and activator of transcription 3 (*STAT3*)/cluster of differentiation 44 (*CD44*) signatures. Elevated mRNA levels of *GSK3β/β-Catenin*/*STAT3*/*CD44* were found to be associated with shorter survival times in patients with glioblastoma multiforme (GBM) (GBM Metabase and SurvExpress). *p* values are indicated in each panel.

**Figure 4 ijms-22-02464-f004:**
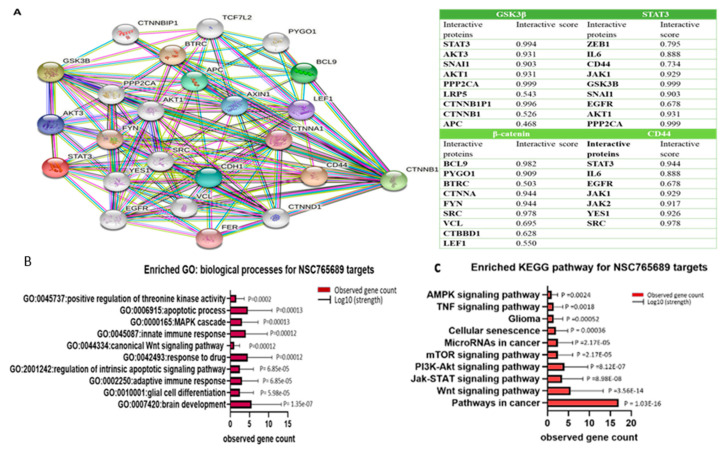
STRING database predicted protein-protein interacting networks (PINs) among the oncogenic markers of glycogen synthase kinase 3β (GSK3B)/*β-Catenin*/signal transduction and activator of transcription 3 (*STAT3*)/cluster of differentiation 44 (*CD44*). (**A**) The network has four initial proteins formed and was further expanded by an additional 30 nodes/protein, by using the “more” key on the STRING interface. Furthermore, a PIN enrichment *p* value of <1.0 × 10^−16^ was obtained from the network, with a clustering coefficient of 0.711. The accompanying table shows all other interacting proteins with *GSK3β*/*β-Catenin*/*STAT3*/*CD44*, and the confidence cutoff value representing interaction links was adjusted to 0.900 as the highest scoring link. (**B**) Biological processes and (**C**) KEGG pathways associated with of *GSK3β*/*β-Catenin*/*STAT3*/*CD44* clustering networks.

**Figure 5 ijms-22-02464-f005:**
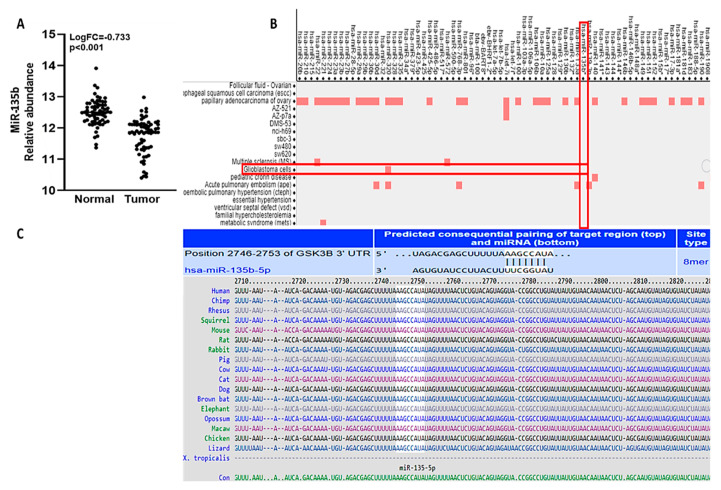
Prediction of expression levels and long-term survival using miRNA-135b-5p and *glycogen synthase kinase 3β* (*GSK3β*) in glioblastoma multiforme (GBM). (**A**) The miRDB prediction tool revealed low expression of *miR-135b* in tumor cohorts compared to adjacent normal tissue samples with *p* < 0.05. (**B**) miRNA-135b-5p expression was low in GBM patients (miRondala-DB). (**C**) Using targetscan online software, we identified the *miR-135 -5p* site on the 3′-untranslated region of *GSK3β mRNA*, which predicted *GSK3β as* a target gene of *miR-135b-5P*.

**Figure 6 ijms-22-02464-f006:**
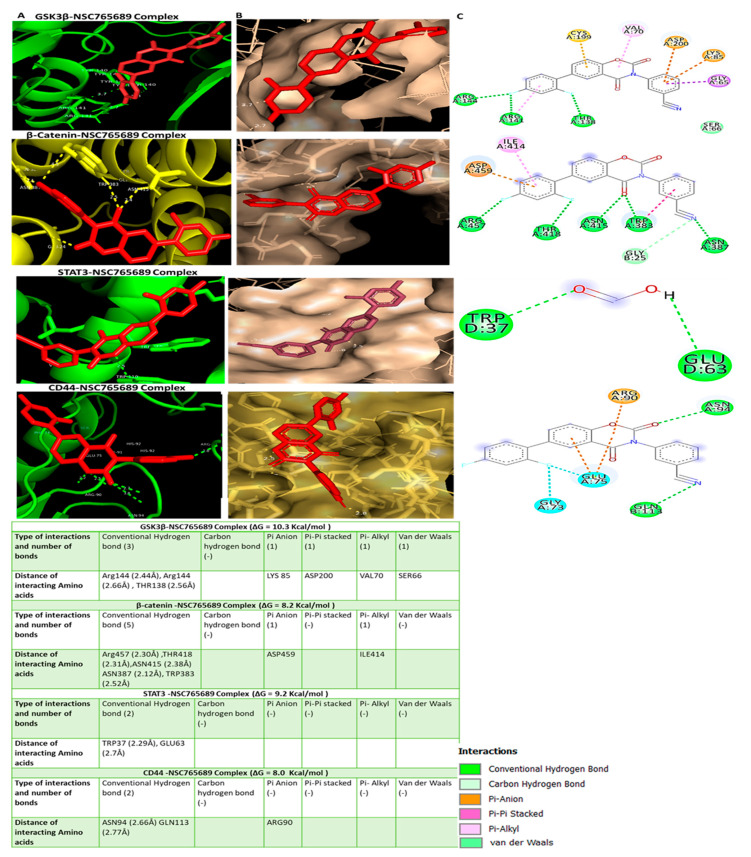
NSC765689 could serve as a potential small-molecule inhibitor of multi-oncogenic proteins. (**A**) 3D structures of *glycogen synthase kinase 3β* (*GSK3β*)/*β-Catenin*/signal transduction and activator of transcription 3 (*STAT3*)/cluster of differentiation 44 (*CD44*)-NSC765689 interactions (left). (**B**) Surface representations of the active-site flap of the *GSK3β*/*β-Catenin*/*STAT3*/*CD44*-NSC765689 complex in the binding pocket (middle). (**C**) 2D structural analysis of amino acids and distances of *GSK3β*/*β-Catenin*/*STAT3*/*CD44*-NSC765689 complexes using Discovery Studio (right). Accompanying Table shows binding affinities, ligand-receptor interactions, atoms involved in H-bonding, bond distances between NSC765689 and target proteins, and interacting amino acid residues.

**Figure 7 ijms-22-02464-f007:**
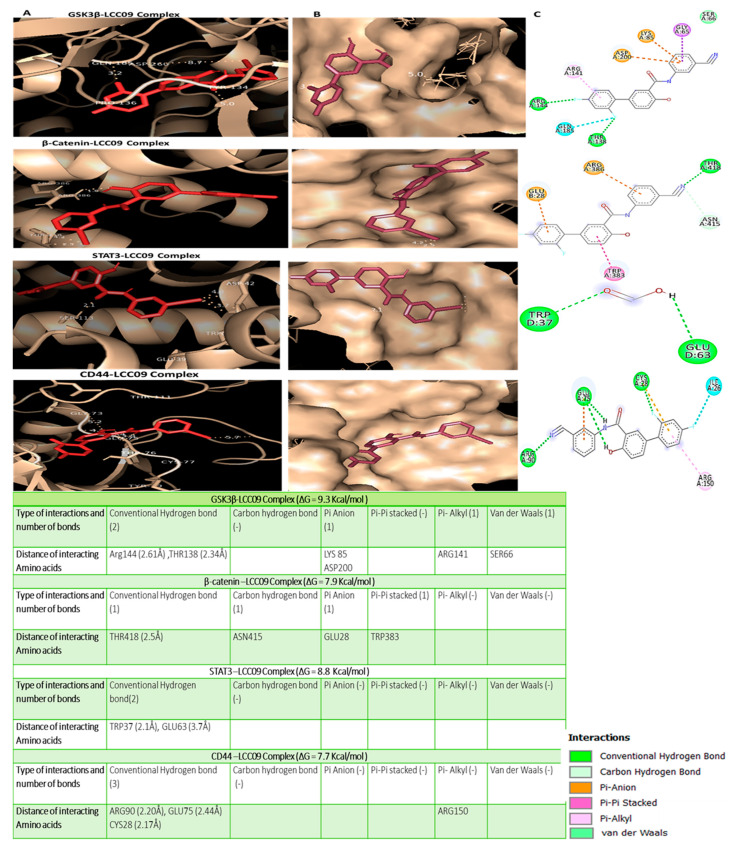
Docking results of LCC09 with *glycogen synthase kinase 3β* (*GSK3β*)/*β-Catenin*/signal transduction and activator of transcription 3 (*STAT3*)/cluster of differentiation 44 (*CD44*). (**A**) 3D structures of *GSK3β/β-Catenin*/*STAT3*/*CD44*-LCC09 interactions (left). (**B**) Surface representations of the active-site flap of the *GSK3β/β-Catenin*/*STAT3*/*CD44*-LCC09 complex in the binding pocket (middle). (**C**) 2D structural analysis of amino acids and distances of the *GSK3β/β-Catenin*/*STAT3*/*CD44*-LCC09 complex using Discovery Studio (right). Accompanying table shows binding affinities, ligand-receptor interactions, atoms involved in H-bond distances between LCC09 and target proteins, and the interacting amino acid residues.

**Figure 8 ijms-22-02464-f008:**
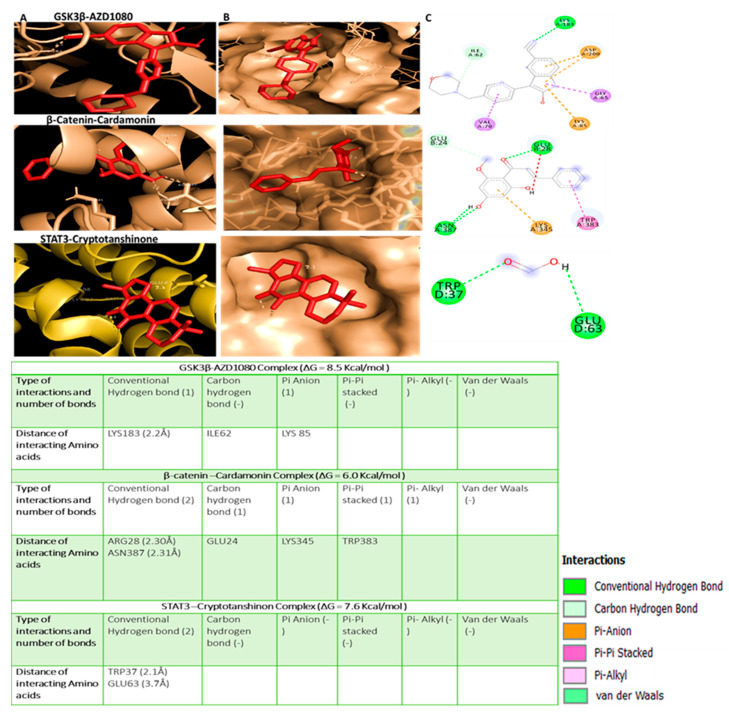
Docking results of AZD1080 with *glycogen synthase kinase 3β* (*GSK3β*)/*β-Catenin*/signal transduction and activator of transcription 3 (*STAT3*). (**A**) 3D structures of *GSK3β*/*β-Catenin*/*STAT3* -AZD1080 interactions (left). (**B**) Surface representations of the active-site flap of the *GSK3β*/*β-Catenin*/*STAT3* -AZD1080 complex in the binding pocket (middle). (**C**) 2D structural analysis of amino acid and distances of the *GSK3β*/*β-Catenin*/*STAT3* -AZD1080 complex using Discovery Studio (right). Accompanying table shows binding affinities, ligand-receptor interactions, atoms involved in H-bonding, bond distances between AZD1080 and target proteins, and interacting amino acid residues.

**Figure 9 ijms-22-02464-f009:**
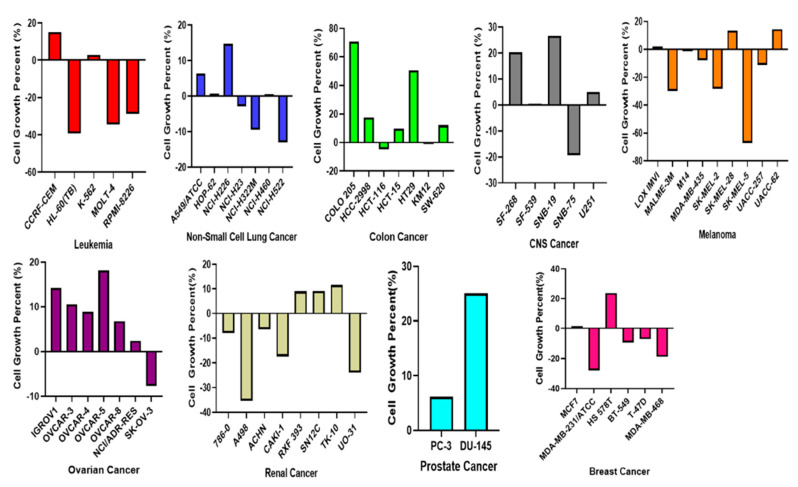
Sensitivity of the NCI-60 cancer cell lines to inhibitory activities of NSC765689. An initial single dose of 10 μM was used on each cell line to test the compound. The zero on the *x*-axis indicates the mean percentage of cell growth of the cell lines. The percentage growth of each cell line relative to the mean is represented by horizontal bars, which are elongated to the right side indicating greater sensitivity and to the left side indicating reduced sensitivity.

**Figure 10 ijms-22-02464-f010:**
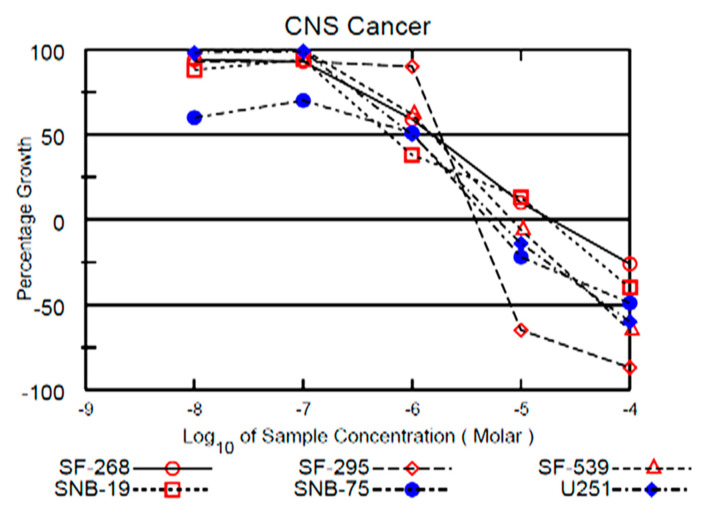
Dose-response curves of the cytotoxic activity of NSC765689 against central nervous system (CNS) cancer cell lines. Growth percentage value of 100 represents the growth of untreated cells, while a value of 0 represents no net growth throughout the period of the experiment, and a value of -100 indicates that all of the cells had been killed by the end of the experiment.

**Table 1 ijms-22-02464-t001:** NCI synthetic compounds and standard anticancer agent sharing similar anti-cancer fingerprints and mechanistic correlation with NSC765689.

			NSC-Synthetic Compounds				NSC-Standard Agents	
Rank	r	CCLC	Target NSC	Target Descriptor	MW	r	CCLC	Target Descriptor
1	0.59	59	757391	DICHLOROPHENE	269.12 g/mol	0.4	57	bryostatin 1
2	0.59	59	29027	TETRACHLOROCATECHOL	247.92 g/mol	0.37	58	4-ipomeanol
3	0.58	59	825282	CPI455	314.77 g/mol	0.36	58	Pentamethyl melamine
4	0.58	56	768421	5-CHLORO-2-(3-(3,4,5-TRIMETHOX	383.01 g/mol	0.33	46	rhizoxin
5	0.58	54	406472	OCTOXYNOL 9(USAN)	250.19 g/mol	0.29	58	gallium nitrate
6	0.57	58	767063	3-(1-(3,4-DICHLOROPHENYL)-2,5-	308.89 g/mol	0.28	58	flavoneacetic acid
7	0.57	59	767322	RO3200934-000	371.41 g/mol	0.27	46	didemnin B
8	0.56	56	768406	5-FLUORO-2-(3-(3,4,5-TRIMETHOX	470.17 g/mol	0.27	58	ARA-6-MP
9	0.56	56	755806	XL147	448.52 g/mol	0.27	56	tamoxifen
10	0.56	56	756322	GW701032X	413.51 g/mol	0.27	54	doxorubicin (Adriamycin)
11	0.56	59	755984	PNU-74654	320.34 g/mol	0.26	57	pibenzimol hydrochloride
12	0.55	56	772889	MK-2048	461.87 g/mol	0.26	46	cyanomorpholino-ADR
13	0.55	56	750742	(E)-3-(6-(4-METHOXYPHENYL)-2-M	177.18 g/mol	0.26	46	S-trityl-L-cysteine
14	0.54	52	328477	BENZAMIDE, N-(3-CHLORO-2-METHY	134.08 g/mol	0.25	58	chromomycin A3
15	0.54	49	401443	PHLORETIN	274.26 g/mol	0.25	58	mitramycin

r—Pearson’s correlation coefficient: value ranges between −1 and 1 (values becomes more significant as they increase above 1), CCLC—common cell lines count cell counts, MW—Molecular weight (g/mol).

**Table 2 ijms-22-02464-t002:** Showing common names, Uniprot and ChEMBL IDs as well as target classes of specific. protein targets of NSC765689.

Target	Common Name	Uniprot ID	ChEMBL ID	Target Class
Cyclin-dependent kinase 9	CDK9	P50750	CHEMBL3116	Kinase
Tyrosine-protein kinase JAK3	JAK3	P52333	CHEMBL2148	Kinase
6-phosphofructo-2-kinase/fructose-2,6- bisphosphatase 3	PFKFB3	Q16875	CHEMBL2331053	Enzyme
Epidermal growth factor receptor erbB1	EGFR	P00533	CHEMBL203	Kinase
Glycogen synthase kinase-3 beta	GSK3B	P49841	CHEMBL262	Kinase
MAP kinase p38 alpha	*MAPK*14	Q16539	CHEMBL260	Kinase
Cyclin-dependent kinase 2	CDK2	P24941	CHEMBL301	Kinase
Cyclin-dependent kinase 1	CDK1	P06493	CHEMBL308	Kinase
Hexokinase type IV	GCK	P35557	CHEMBL3820	Enzyme
Tankyrase-2	TNKS2	Q9H2K2	CHEMBL6154	Enzyme
Tankyrase-1	TNKS	O95271	CHEMBL6164	Enzyme
Fructose-1,6- bisphosphatase	FBP1	P09467	CHEMBL3975	Enzyme
11-beta-hydroxysteroid	HSD11B1	P28845	CHEMBL4235	Enzyme
Poly [ADP-ribose]	PARP1	P09874	CHEMBL3105	Enzyme
Nicotinamide	NAMPT	P43490	CHEMBL1744525	Enzyme
Signal transducer and activator of transcription 3	*STAT3*	P40763	CHEMBL4026	Transcription factor
PI3-kinase p110-delta subunit	PIK3CD	O00329	CHEMBL3130	Enzyme
Thrombin	F2	P00734	CHEMBL204	Protease

**Table 3 ijms-22-02464-t003:** Cytotoxic activities of NSC765689 against 60 NCI human cancer cell lines. GI_50_, 50% growth inhibition; TGI, total growth inhibition; LC_50_, 50% loss of cells.

Panel/Cell line (μM)	NSC765689		
	GI50	TGI	LC50
Leukemia			
CCRF-CEM	0.65	20.86	15.70
HL-60(TB)	0.52	10.80	>100
K-562	2.40	58.2	>100
MOLT-4	0.30	17.7	43.90
RPMI-8226	0.69	38.0	47.90
Non-Small Cell Lung Cancer			
A549/ATCC	1.87	10.80	>100
EKVX	1.88	6.94	96.20
HOP-62	1.24	0.67	>100
HOP-92	0.55	5.48	>100
NCI-H226	0.41	2.77	53.40
NCI-H23	1.80	8.91	>100
NCU-H322M	0.85	12.30	>100
NCI-H46-	0.36	2.31	43.40
Colon cancer			
COLON 205	5.13	17.90	53.30
HCC2998	2.88	7.91	50.40
HCT-116	1.14	6.01	38.80
HCT-15	1.20	10.70	>100
HT-29	4.05	15.10	>100
KM12	1.73	7.31	>100
SW-620	2.08	31.10	<100
CNS cancer			
SF-295	1.53	18.90	8.01
SF-539	1.81	3.81	55.50
SNB-19	1.51	8.06	>100
SNB-75	0.61	17.40	>100
U251	1.03	4.99	61.40
Melanoma			
LOX IMVI	1.23	3.91	8.26
MALME-3M	0.38	2.71	>100
M14	0.24	9.96	>100
MDA-MB-435	1.68	5.13	>100
SK-MEL-2	0.39	1.83	7.81
SK-MEL-28	0.43	4.76	>100
SK-MEL-5	0.23	0.10	3.08
UACC-257	0.73	3.71	>100
UACC-62	0.36	1.91	35.80
Ovarian cancer			
IGROV1	0.88	21.50	>100
OVCAR-3	0.55	12.50	>100
OVCAR-4	0.38	5.74	>100
OVCAR-5	4.76	100.00	>100
OVCAR-8	1.77	12.00	>100
NCI/ADR-RES	0.54	1.00	>100
SK-OV-3	0.75	0.70	>100
Renal cancer			
786-0	2.94	2.94	41.40
A489	0.40	0.40	15.80
ACHN	1.18	1.18	62.00
CAKI-1	2.35	2.35	>100
RXF 393	1.10	1.10	92.30
SN12C	0.92	0.92	>100
TK-10	2.89	2.89	>100
UO-31	1.62	1.62	>100
Prostate cancer			
PC-3	1.14	1.14	>100
DU-145	2.57	2.57	>100
Breast cancer			
MCF7	0.36	4.26	>100
MDA-MB-231/ATCC	1.28	4.89	68.10
HS 578-T	0.66	5.28	>100
BT-549	0.29	0.99	>100
T-47D	0.37	2.88	>100
MDA-MB-468	1.17	3.21	>100

## Data Availability

The datasets generated and/or analyzed in this study are available on reasonable request.

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
