# Peer review of "An Integrated Bioinformatics Study of a Novel Niclosamide Derivative, NSC765689, a Potential GSK3β/β-Catenin/STAT3/CD44 Suppressor with Anti-Glioblastoma Properties"

_ijms, 2021, doi:10.3390/ijms22052464_

Round 1
Reviewer 1 Report
Re: Manuscript ID: ijms-1106197 / IJMS (ISSN 1422-0067)
Type of manuscript: Article
Title: An Integrated Bioinformatics Study of a Novel Niclosamide Derivative, NSC765689, a potential GSK3β/β-Catenin/STAT3/CD44 suppressor with Anti-Glioblastoma Properties
Authors
Ntlotlang Mokgautsi, Ya-Ting Wen, Bashir Lawal, Harshita Khedkar, Maryam Rachmawati Sumitra, Alexander T.H. Wu, Hsu-Shan Huang
In this present study, Mokgautsi et al. used random computer-based prediction and target identification to assess activities of a synthesized niclosamide-derived compound, NSC765689, to target GBM oncogenic signaling. Using the online algorithms they identified glycogen synthase kinase 3β, β-catenin, signal transducer and activator of transcription 3, and cluster of differentiation 44 as potential druggable candidates of NSC765689. They also identified microRNA (miR)-135b as one of the most suppressed microRNAs in GBM samples using bioinformatics, which was reported to be upregulated through inhibition of GSK3β, and subsequently suppresses GBM tumorigenic properties and stemness. They further performed in silico molecular docking of NSC765689 with GBM oncogenes; GSK3β, β-catenin, and STAT3, and the stem cell marker, CD44, to predict protein-ligand interactions. The results indicated that NSC765689 exhibited stronger binding affinities compared to its predecessor, LCC09, and was proven to inhibit GBM stemness and resistance. They then used available US National Cancer Institute 60 human tumor cell lines service to screen in vitro anticancer effects, and concluded that further in vitro and in vivo investigations should be performed to validate the efficacy of NSC765689 as a novel potential GBM therapeutic.
The results reported represent a notable advance in search of novel anti-GBM drugs. In general, I found the manuscript is well written and structured smoothly. The bioinformatic tolls are technically sound and scientifically valid. Detailed and systematic studies are presented. In my opinion, the manuscript is suitable for publication in IJMS, after the authors have addressed my comments and questions.
- The manuscript presents only results obtained through web tolls and contract work. No experimental results are available on the validation of those claimed results, which is perhaps more interesting in terms of potential applications. I understand that some difficulties may prevent the authors to achieve the measurement. Can the authors see any indications in gene and/or expression? From the context, this lab did wet lab work confirming their discoveries regarding this compound. It will be helpful if the authors can provide some experimental information in this current manuscript.
- Even though there are important advances in the design, the analysis and results are repetition of previous work. Not much new biology was learned. The analysis was repeating what has been done in many previous papers. Given this fact, the analysis was also far too lengthy for a compact journal, as if it was carrying an aspect of novelty. The discussion should be organized more systemically with more clear statement.
Author Response
Responses to Check Results
Journal Name: International Journal of Molecular Sciences (IJMS).
Manuscript Title: Title: An Integrated Bioinformatics Study of a Novel Niclosamide Derivative, NSC765689, a potential GSK3β/β-Catenin/STAT3/CD44 suppressor with Anti-Glioblastoma Properties
Dear Editor,
Thank you for your comments and suggestions on my manuscript to this highly reputable journal. The manuscript has been extensively revised based on your advice and the comments and suggestions of the referees. We have modified the manuscript accordingly, and detailed corrections are listed below point by point:
Comments:
Point 1: The manuscript presents only results obtained through web tolls and contract work. No experimental results are available on the validation of those claimed results, which is perhaps more interesting in terms of potential applications. I understand that some difficulties may prevent the authors to achieve the measurement. Can the authors see any indications in gene and/or expression? From the context, this lab did wet lab work confirming their discoveries regarding this compound. It will be helpful if the authors can provide some experimental information in this current manuscript.
Response 1: In this current study, we focused mainly on computer-based predictions and target identification, which revealed that our compound NSC765689 may be a lead compound, which exhibits antipro liferative and cytotoxic properties in glioblastoma multiforme (GBM). The compound showed potential to regulate GBM signaling GSK3β, β-catenin, STAT3, and CD44. Therefore, here we mainly focused on predictions as proof of concept, in addition to this, we recently published NSC765689 predecessor (Cancers 2019, 11, 1442; doi:10.3390/cancers11101442), LCC09 (open ring) which proved to inhibit GBM stemness and resistance both invitro and invivo, and according to our in silico predictions NSC765689 (closed ring) proves to have better activities as compared to LCC09. This study is part 1 of 2, further validation in vitro and in vivo studies in tumor bearing mice to assess the compound’s full therapeutic efficacy are in progress and will be prepared in next coming manuscript soon.
Point 2: Even though there are important advances in the design, the analysis and results are repetition of previous work. Not much new biology was learned. The analysis was repeating what has been done in many previous papers. Given this fact, the analysis was also far too lengthy for a compact journal, as if it was carrying an aspect of novelty. The discussion should be organized more systemically with more clear statement.
Response 2: We have revised the discussions and organized it more systemically, by rearranging our results figures, and therefore making our statement clearer.
- Finally, as advised we have outlined above, a point-by-point response to each of the referee’s comments with cogent scientific explanations. We hope that the revised manuscript, after the incorporation of these changes, would now be formally accepted in the IJMS. We look forward to your positive response.
Reviewer 2 Report
The authors characterize a novel compound, NSC765689, as a potential candidate for GBM treatment. The authors demonstrate that NSC765689 meets the criteria of a small drug candidate, and that it targets, among others, GSK3β, β-catenin, STAT3, and CD44 proteins associated with stemness and EMT in GBM and positively correlated with poor prognosis. Overall, the paper demonstrates with robust in silico experiments the potential of targeting the mentioned proteins in GBM, and show the ability of NSC765689 to inhibit GBM cells’ growth in vitro at moderate concentrations. In my opinion, it merits publication. I have some minor comments and suggestions that might improve the quality of the article.
- Some figures’ quality was compromised by distortive disproportionate rescaling. In particular, the Figure 3a is hard to interpret because of the background.
- It would be interesting to observe experiments demonstrating that the toxicity of NSC765689 in GBM cells is owed to the inhibition of the suggested proteins. I suggest to perform WB assays to demonstrate the inhibition proteins downstream on the Wnt/β-catenin pathway or the STAT3 downstream signaling. It would also improve the quality of the study to demonstrate the upregulation of miR-135b in response to NSC765689 in GBM cells.
- A mention of the potential of NSC765689 as an inhibitor for the other protein targets (such as CDK9 and PARP) would be of interest for cancer researchers.
Author Response
Responses to Check Results
Journal Name: International Journal of Molecular Sciences (IJMS).
Manuscript Title: Title: An Integrated Bioinformatics Study of a Novel Niclosamide Derivative, NSC765689, a potential GSK3β/β-Catenin/STAT3/CD44 suppressor with Anti-Glioblastoma Properties
Dear Editor,
Thank you for your comments and suggestions on my manuscript to this highly reputable journal. The manuscript has been extensively revised based on your advice and the comments and suggestions of the referees. We have modified the manuscript accordingly, and detailed corrections are listed below point by point:
Comments:
Point 1: Some figures’ quality was compromised by distortive disproportionate rescaling. In particular, the Figure 3a is hard to interpret because of the background.
Response 1: We have replaced the distorted figure 3a with an improved quality figure.
Point 1: It would be interesting to observe experiments demonstrating that the toxicity of NSC765689 in GBM cells is owed to the inhibition of the suggested proteins. I suggest to perform WB assays to demonstrate the inhibition proteins downstream on the Wnt/β-catenin pathway or the STAT3 downstream signaling. It would also improve the quality of the study to demonstrate the upregulation of miR-135b in response to NSC765689 in GBM cells.
Response 1: In this current study, we focused mainly on computer-based predictions and target identification, which revealed that our compound NSC765689 may be a lead compound, which exhibits antipro liferative and cytotoxic properties in glioblastoma multiforme (GBM). The compound showed potential to regulate GBM signaling GSK3β, β-catenin, STAT3, and CD44. Therefore, here we mainly focused on predictions as proof of concept, in addition to this, we recently published NSC765689 predecessor (Cancers 2019, 11, 1442; doi:10.3390/cancers11101442), LCC09 (open ring) which proved to inhibit GBM stemness and resistance both invitro and invivo, and according to our in silico predictions NSC765689 (closed ring) proves to have better activities as compared to LCC09. This study is part 1 of 2, further validation in vitro and in vivo studies in tumor bearing mice to assess the compound’s full therapeutic efficacy are in progress and will be prepared in next coming manuscript soon
- Finally, as advised we have outlined above, a point-by-point response to each of the referee’s comments with cogent scientific explanations. We hope that the revised manuscript, after the incorporation of these changes, would now be formally accepted in the IJMS. We look forward to your positive response.